# UniTVelo: temporally unified RNA velocity reinforces single-cell trajectory inference

Mingze Gao [1], Chen Qiao [1] & Yuanhua Huang [1,2] ✉

The recent breakthrough of single-cell RNA velocity methods brings attractive promises to reveal directed trajectory on cell differentiation, states transition and response to perturbations. However, the existing RNA velocity methods are often found to return erroneous results, partly due to model violation or lack of temporal regularization. Here, we present UniTVelo, a statistical framework of RNA velocity that models the dynamics of spliced and unspliced RNAs via flexible transcription activities. Uniquely, it also supports the inference of a unified latent time across the transcriptome. With ten datasets, we demonstrate that UniTVelo returns the expected trajectory in different biological systems, including hematopoietic differentiation and those even with weak kinetics or complex branches.

Single-cell RNA sequencing (scRNA-seq) has already transformed how dynamic biological processes be studied at cellular level. It has enabled the tracking of developmental stages of distinct cell lineages[1,2]. This technique is often referred to as trajectory inference, a set of computational algorithms to infer the order and pseudotime of individual cells along differentiation trajectories[3]. Plenty of methods have been developed for this purpose to model the progression of cells from transcriptome-derived manifolds, with either continuous pseudotime or topologies covering from linear, bifurcation to graph[4–8]. However, since scRNA-seq only captures a static snapshot of the transcriptome of a cell population, most conventional trajectory inference methods lack the ability to automatically identify the direction of the returned trajectory. Hence, these methods often require additional inputs or prior knowledge to specify progenitor cells and differentiated cells[3,9], which consequently limits their applicability for biological processes with unknown cell fate or in abnormal conditions.

On the other hand, the short-term change of expression levels (often referring to the spliced mature RNAs) can be indicated by the commonly captured nascent RNAs (i.e., unspliced RNAs), which also reflect the regulatory activity of transcription[10]. By leveraging the balance between unspliced and spliced mRNA reads during transcription, RNA velocity[11] has further extended the descriptive trajectory model to a predictive manner, where a positive velocity represents a gene being up-regulating whilst a negative velocity stands for down-regulating. Assuming transcription phases last sufficiently long to reach a new equilibrium, La Manno et al formed the fundamental 'steady-state' model of RNA velocity method and consequently projected cells' new states by aggregating velocities across all genes[11]. Recently, Bergen and colleagues further extended the RNA velocity quantification and introduced the scVelo package which contains the covariance-based 'stochastic' mode and the likelihood-based 'dynamical' mode[12].

However, velocity estimations are still found to be inaccurate or inconsistent when recovering cellular transitions[13,14], partly because of the severely low signal-to-noise ratio in unspliced mRNAs. More importantly, current models either rely on linear assumptions to form a steady-state regression line or presume a time-invariant transcription rate for a certain state (e.g., induction, repression or both phases), which are often violated and may result in distorted or even reversed velocity matrix estimation, e.g., in ref. 15. Possible solutions include manually removing MUltiple Rate Kinetics (MURK) genes that violates model assumption[15], identifying differential momentum genes[16], projecting high dimensional transcriptomics onto effective embeddings[17], or enriching the nascent RNAs via metabolic labeling techniques[13,18].

Here, to circumvent the limitations of linear assumptions that a gene ought to exhibit dynamical traits and instead of focusing on the refinement of pre- or post-processing modules, we focused on the core velocity estimation step and developed UniTVelo, a statistical method that models the full dynamics of gene expression with a radial basis function (RBF) and quantifies RNA velocity in a top-down manner. Uniquely, we also introduce a unified latent time across the whole transcriptome, which can resolve the discrepancy of directionality

[1]School of Biomedical Sciences, University of Hong Kong, Hong Kong SAR, China. [2]Department of Statistics and Actuarial Science, University of Hong Kong, Hong Kong SAR, China. ✉e-mail: yuanhua@hku.hk

between genes. The capabilities and generalization abilities of Uni-TVelo is demonstrated on various developmental trajectories across 10 datasets (Supplementary Table S1), each with its unique feature, including erythroid haematopoiesis lineages[19,20] which contain MURK genes, retina development with a clear cell cycle phase[21] and multi-branching scenario in bone marrow differentiation[22].

## Results

### High-level description of UniTVelo model

Same as previous frameworks for RNA velocity quantification[11,12], we formulate the transcription activity and splicing kinetics for each gene independently by a linear first-order dynamic system (Fig. 1a),

$$
\begin{aligned}
\frac{du(t)}{dt} &= \alpha(t) - \beta u(t) \\
\frac{ds(t)}{dt} &= \beta u(t) - \gamma s(t),
\end{aligned}
\tag{1}
$$

where $u(t)$, $s(t)$ respectively represents normalized unspliced and spliced mRNA reads, with full transcription dynamics being described in a temporal relationship by transcription rate $\alpha(t)$, splicing rate $\beta$ and degradation rate $\gamma$ (Fig. 1b).

Here, we innovated the RNA velocity quantification in two folds. First of all, we introduced a spliced RNA-oriented design to model the RNA velocity and transcription rate functions. Previous methods, including both velocyto and scVelo, determine the gene expression dynamics along with the order of data generation. Namely, the transcription rate $\alpha(t)$ is first defined, commonly with a step function, followed by deriving the profiles of unspliced and spliced RNAs via Eq.(1) (Supplementary Fig. S1a). Differently, our proposed UniTVelo utilizes a top-down strategy by directly designing a profile function of

spliced RNAs $s(t) := f(t; \boldsymbol{\theta})$, where $\boldsymbol{\theta}$ is a set of gene-specific parameters controlling the shape of phase portraits, then derives the dynamics of unspliced RNAs and transcription rates also via Eq.(1) (Fig. 1b; Methods). In principle, the gene expression dynamical function $f$ can be broadly chosen, e.g., sigmoid or bell-shape functions or complex neural network models. Here, we chose a radial basis function (RBF; Methods), which has been validated its usefulness in modeling transcriptome dynamics[23] and has the ability to capture the induction, repression, and transient shapes with a single function family (Fig. 1c and Methods).

Intrinsically, this top-down design allows more flexible gene expression profiles than that derived from the step function of transcription rates (two-state burst model, e.g., scVelo), which therefore may mitigate model violation in complex transcription regulations[15,24]. On the other hand, this reversed design remains highly capable of fitting data generated in a forward manner, i.e., via a step function of transcription rate (Supplementary Fig. S1c), suggesting high robustness of this framework in quantifying RNA velocity, even with data generated from a different model. Furthermore, UniTVelo's top-down framework with RBF is able to retain a similar but smooth relation between spliced and unspliced RNAs compared to that of a bottom-up framework in scVelo with regards to dynamical genes (Fig. 1d; Supplementary Fig. S1d). This comparable performance is further evidenced in a systematic comparison of the entire transcriptome on a per gene basis (Supplementary Fig. S2; Results).

Thanks to the spliced RNA oriented design, the velocity of each gene can be obtained directly from the derivative of spliced mRNA function $f$, rather than the deviation to steady-state equilibrium (Methods). This benefits the RNA velocity quantification based on more reliable spliced RNAs and allows cells that are either above or

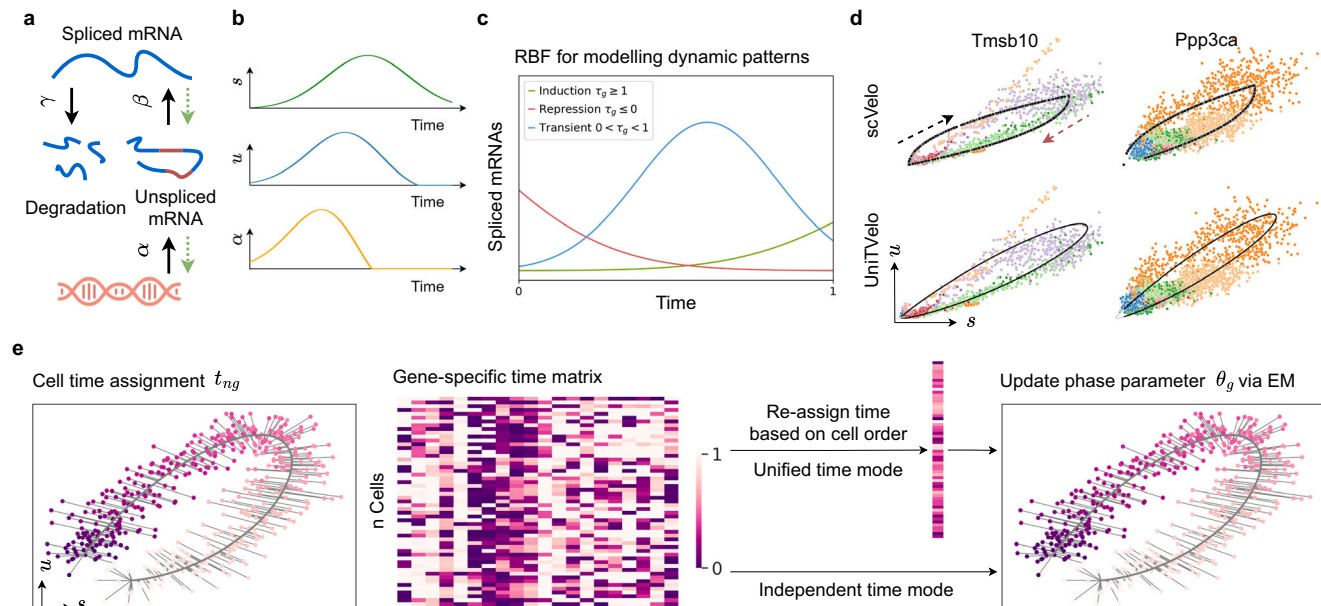

**Fig. 1 | Illustration of UniTVelo for modelling of transcription dynamics and RNA velocity. a** Illustration of transcriptional process which involves transcription rate $\alpha$, splicing rate $\beta$, and degradation rate $\gamma$. Green dotted line indicates that parameters are inferred reversely. **b** Paradigm of the model in **a** with time as independent variable, showing predicted changes of $\alpha$ with regards to measured expression profile. **c** RBF for modeling patterns of induction, repression, and transient dynamics of each gene, where $\tau_g$ represents the peak time. Latent time of this model is rescaled and truncated between 0 and 1. **d** Example of phase portraits (can be splitted to induction and repression stages from the middle, shown in black and red arrows respectively) of two dynamical genes modeled by both scVelo and UniTVelo, *Tmsb10* and *Ppp3ca*, showing RBF kernel has a similar ability to recover

gene's dynamic information. Colors indicate various cell types. **e** The inference of UniTVelo which tries to recover genes' dynamic process via two sets of parameters: (1) gene-specific parameters $\theta_g$ which define how transcriptome of each gene changes along time and the relationship between un/spliced mRNA in progression. (2) cell-specific time points $t_{ng}$. By iteratively updating the gene-specific parameters using gradient descent, cell time points are assigned by minimizing the euclidean distance to phase trajectory. Specifically, besides directly using gene-specific time matrix in optimization, UniTVelo also supports a unified-time assignment for each cell based on cell ordering. This enables the discrepancy between genes' directionality to be minimized.

under the steady-state equilibrium to be both assigned in steady-state, instead of forcibly dividing into induction and repression stages as done by previous framework. Overall, this top-down framework enjoys computational convenience of both modelling the spliced RNAs with diverse distribution families, including deep neural networks as demonstrated in ScTour[25] and aggregating latent time across genes (see below), while preserving the same level of accuracy in a vanilla setting.

Our second major innovation is the introduction of a unified latent time across the whole transcriptome when inferring gene expression dynamics and RNA velocity (Fig. 1e; Methods). Conventionally, a framework like scVelo fits the phase portrait to an almond shape on each gene individually, which could easily over-fit due to the high technical noise and the complexity of genes' activities. By contrast, this time-unified setting in UniTVelo can aggregate the dynamic information across all genes to further reinforce the temporal ordering of cells (demonstrated below). This is critically important, as it allows to effectively incorporate stably and monotonically changed genes, namely, their expressions change along with time but are within a sub-steady-state continuously (Supplementary Fig. S1e).

A maximum likelihood estimation of UniTVelo model is achieved by a principled Expectation-Maximization (EM) algorithm (Methods). Briefly, the predicted time for each cell along the differentiation path is updated concurrently during optimizing the parameters of the dynamical system (Fig. 1e). Furthermore, we retain two distinct modes in UniTVelo for allocating the latent time to account for various types of lineage datasets (Fig. 1e; Methods). Specifically, unified-time mode of the algorithm (the default setting) is designed to address the scenario when genes rarely have classic dynamic traits whilst independent mode of UniTVelo is intended for more complicated datasets, for instance, datasets with cell cycle or sparse cell types included. This thereby could calculate the velocity of each individual gene in a more precise way.

### Incorporating multi-rate kinetic genes and revealing erythroid maturation

To evaluate our UniTVelo, we first applied it on both mouse and human erythroid progression datasets, where a set of genes were found with multiple rate kinetics (MURK)[15] hence violating the conventional model assumption and limiting the applicability of current RNA velocity methods[14]. By re-analyzing with scVelo, we also evidenced the distorted lineage inference. In contrast, by using a unified time, UniTVelo corrects the trajectory directions to the expected erythroid maturation on both datasets (9k mouse cells from Blood progenitors 1 to Erythroid 3[19], and 30k human cells from MEMP, Megakaryocyte-Erythroid-Mast cell Progenitor, to Late Erythroid[20]; Fig. 2a, b). Both velocity streams and latent time assignments demonstrates the correctness of UniTVelo (Supplementary Fig. S3a, b).

Further, in the mouse dataset we looked into the MURK genes that were identified in ref. 15, e.g. *Abcg2* and *Smim1* (Fig. 2c). Probably due to the transcription boosting in later differentiation stages, the gene-independent mode in scVelo returns erroneous directions representing a repression shape, even though it fits the data more tightly. By contrast, UniTVelo rescues the direction on these genes by jointly fitting all genes along with a unified time. Additionally, beside MURK genes, a large proportion of genes in erythroid lineage exhibits obvious dynamic information whilst provides vague directionality information during differentiation process (e.g. gene *Cnn3* and *Cyr61* in Fig. 2d), and as a result, cells are scattered around the steady-state regression line with no obvious deviation. We argue that this group of genes can easily lead to over-fitting in a gene-independent setting (as produced by scVelo; Fig. 2c,d). On the other hand, these genes are dynamic informed, hence can strengthen the smoothness of the trajectory if correcting the directionality by a unified time setting in UniTVelo (Fig. 2c, d).

Additionally, thanks to the spliced RNA oriented design, the $\tau_g$ parameter in the RBF denotes the time when gene $g$ reaches its highest expression, hence indicating the profile pattern of each gene in a predefined time window (0 to 1; see Fig. 1c). In the mouse dataset, we found a majority of genes have $\tau_g < 0$, suggesting a repression pattern, while a smaller set of genes have $0 < \tau_g < 1$ for transient or $\tau_g > 1$ for induction patterns (Fig. 2e). By stratifying genes with the $\tau_g$ parameter, we evidenced their expressions along the inferred time indeed match the anticipated shape (Fig. 2f; Supplementary Fig. S3c). As an example, the predicted transient gene *Scube2* did not immediately show a strong transient pattern in the phase portrait for itself alone (Fig. 2g), probably due to high technical noise. Interestingly, when visualizing the dynamics of spliced and unspliced RNAs separately, the bell-shape transient patterns are clearly displayed (Fig. 2g), suggesting that the unified time can further capture the transient pattern on top of recovering the expected cell ordering.

### Identifying the progression kinetics and direction in bone marrow development

Next, we asked if UniTVelo is applicable to reveal cell differentiation on a dataset of full bone marrow development[22], which have complex progressions from hematopoietic stem cells (HSCs) to three distinct branches: erythroids, monocytes, and common lymphoid progenitors (CLP). When re-analyzing it with scVelo, we found it again returns reversed direction on the erythroid branch, similar to reports from the original authors[14], and also distorted trajectory on the monocyte branch (Fig. 3b; Supplementary Fig. S4a for stochastic mode; Supplementary Fig. S4b for latent time). As a comparison, UniTVelo recovers the corrected velocity direction on both scenarios through using a unified time (Fig. 3a), which is in good concordance with the pseudotime inferred by a diffusion map method with starting node specified manually[22].

Besides the global perspective of progression directionality, UniTVelo also facilities the analysis of gene's phase portraits (Fig. 3c). Due to the complexity of the gene's multi-kinetic behavior in micro-environments, the regression function needs to be flexible accordingly. Three clear examples are *Cd44*, *Celf2,* and *Taok3* which are overall highly expressed in the process and related to cell adhesion or cancer progression. It has been shown that spliced mRNA counts of these genes are decreasing gradually as the differentiation approaches terminal states, though an elevated unspliced counts expression is observed in *Cd44* and *Celf2* surprisingly. UniTVelo correctly identified the underlying biological process together with putative time information, suggesting that using a unified time during optimization contributes to the overall performance (Supplementary Fig. S3c, d). UniTVelo, again, concurrently classifies genes into three categories, induced in temporal space, repressed or transient, differing in the peak time (or inferred cell time) of expression profile, as also confirmed via heatmaps (Fig. 3d).

### Resolving cell fate transitions in intestinal organoid differentiation

To further evaluate the capability of UniTVelo in resolving cell fate, we applied it to a bifurcating dataset of intestinal organoid differentiation[26] (Fig. 4a). In the original study, metabolic labeling was used together with sequencing information to investigate mRNA control strategies. Two unique differentiation branches were identified by Monocle2[27] and directions were determined manually by annotating stem cells, differentiated cell types in secretory lineage and enterocyte lineage, respectively. To mimic common scRNA-seq without metabolic labeling, we then applied the RNA velocity methods using only unspliced and spliced mRNA counts. scVelo could not correctly capture the entire dynamics along branches which appears as 'split-up' at the middle of enterocyte lineage and has minor disarray in the secretory lineage (Fig. 4b). As a comparison, UniTVelo depicts a

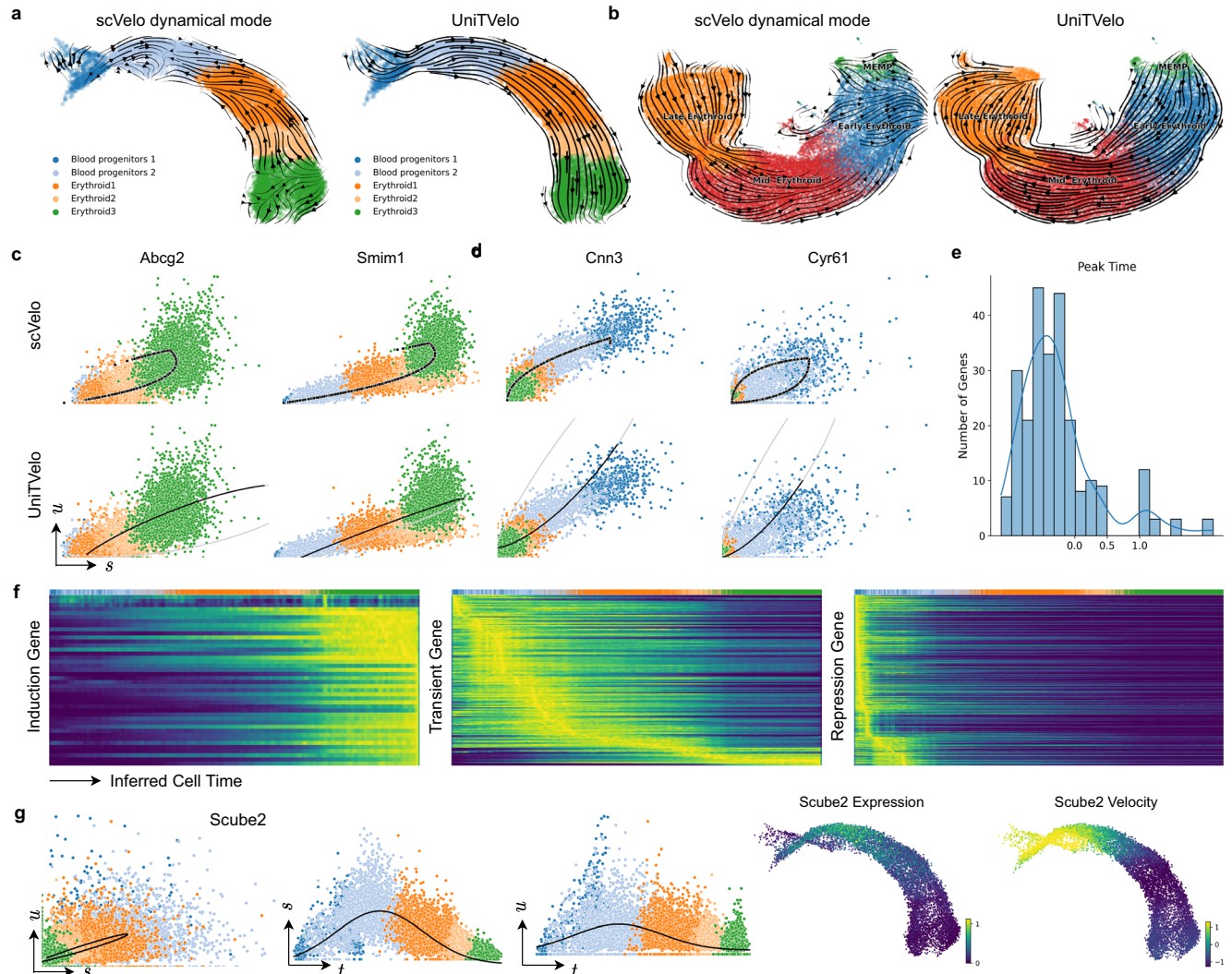

**Fig. 2 | UniTVelo correctly identifies trajectory of both mouse and human erythroid haematopoiesis lineages. a** Velocities derived from scVelo's dynamical mode (left) and UniTVelo (right) of mouse erythroid lineage. **b** RNA velocity is also tested on a human erythroid dataset starting from progenitor cells, illustrating scVelo's dynamical mode (left) could not find the correct directionality compared with UniTVelo. **c**, **d** Four example genes are selected from the mouse erythroid dataset to demonstrate phase portraits depicted from both methods. *Abcg2* and *Simi1* are induction genes whilst *Cnn3* and *Cyr61* are repression genes, the former two are also considered as MURK genes with transcriptional boosting. Upper panel: scVelo. Lower panel: UniTVelo, both gray and black lines are part of phase portraits whilst only black part is used by the model. The direction of the curve is the same as Fig. 1d. **e** A histogram showing the distribution of peak time of each gene in the mouse erythroid dataset, indicating a large proportion of genes' activity is inhibited during differentiation. **f** Genes can be coarsely classified into three types according to peak time. Heatmap along with inferred cell time shows a clear and accurate separation between each type, e.g., induction genes tend to be active at the end of cellular process whilst repression genes behave oppositely. **g** An example of transient genes, *Scube2*, shows misleading time assignments if only one phase portrait is used. However, unified time suggests a clear transient pattern for both unspliced and spliced counts.

clearer and more logical velocity field. Interestingly, it also has the ability to validate local regions, for example, paneth cells are pointing towards goblet cells (Supplementary Fig. S5a), consistent with earlier report on the secretory lineage[26].

Additionally, to assess the goodness-of-fit for each gene, we employed the coefficient of determination $R^2$ (commonly used in regression analysis), namely the mean squared error divided by the variance of spliced mRNA counts (Fig. 4c). This metric not only examines how well the model captures the gene expression profiles but also indicates the informative genes that explain the inferred cell trajectory. Figure 4d shows the phase portraits of a few example genes selected from either higher $R^2$ or lower $R^2$, illustrating that the former scenario tends to exhibit clear dynamic or marker gene characteristics whilst for the latter, the expression profile of unspliced and spliced counts appear to be more stochastic or less abundant. Similarly, these differences can be validated via the

visualized scatter plot of expression profiles (Fig. 4e, Supplementary Fig. S5b).

## Delineating complicated biological systems with independent mode

Besides the default unified-time mode, UniTVelo also retains an independent mode (Methods), similar to scVelo where each gene is analyzed independently and has its own latent time. This flexible setting is useful for datasets with high signal-to-noise ratio, especially for complex differentiation scenarios containing cell cycles or sparse cell types which hampers the performance of unified-time mode (Methods; Supplementary Table S2).

We first validated UniTVelo's independent mode on a neurogenesis dataset[27] in which mouse dentate gyrus was sequenced and measured at two time points (P12 and P35). By large, both scVelo and UniTVelo(independent mode) successfully identified the major

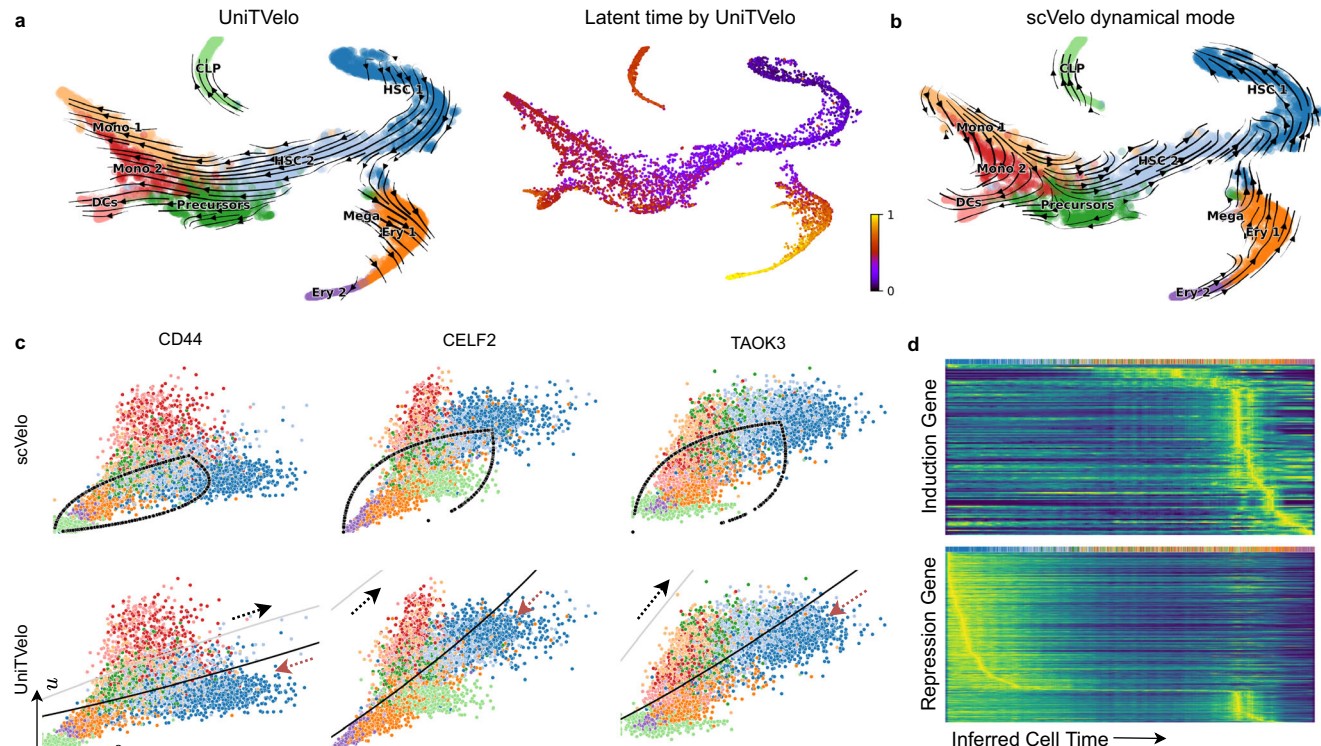

**Fig. 3 | UniTVelo correctly identifies differentiation trajectory of human bone marrow development. a** UMAP coordinates of the velocity field shown in streamlines (left) and predicted latent time (right) from UniTVelo. **b** Estimated RNA velocity field in streamlines by scVelo. **c** *Cd44*, *Celf2,* and *Taok3* are selected as examples, illustrating UniTVelo could accurately capture cell states and directionality whilst scVelo failed to capture the relative cell states using almond shape phase portraits. Upper panel: Regression result by scVelo. Lower panel: Inferred cell state by UniTVelo, gray and black line demonstrates part of induction and repression phase of RBF kernel respectively and the black part is used by the model. Red arrows show the correct directionality of cellular dynamics. **d** Heatmaps of predicted induction and repression gene expressions are resolved along the inferred cell time, showing a clear separation in temporal space. Entities are smoothed spliced counts.

differentiation trajectory in this experiment that neuroblast cells gradually become granule cells (Supplementary Fig. S6a), probably thanks to the similar settings with gene-independent time. On the other hand, UniTVelo returns stronger signals in sub-lineages, especially from oligodendrocyte precursor cells (OPCs) to oligodendrocytes (OLs) as terminal, despite the low number of cells positions, a technical challenge on this branch.

To validate UniTVelo's ability on cycling progenitors and multi-branching scenarios, a mouse retinal development dataset sampled at E15.5 from Lo Giudice et al.[21] is presented. Genes' expression profile and manual annotation reveals cell proliferation cycle exists (Supplementary Fig. S6b) before differentiated into three terminal states, photoreceptors (PR), retinal ganglion cells (RGC), and amacrine/horizontal cells (AC/HC). Interestingly, both scVelo (dynamical mode) and UniTVelo (independent mode) returned strong cycling transition among the progenitor cells, and identified all the three differentiation branches. In detail, scVelo returns a disconnected path in the terminal area of RGC branch, whilst UniTVelo avoids the local disturbances along the trajectory.

Taken together, when the dynamic signal is rich and the setting of gene-independent time is applicable, UniTVelo works similarly well compared to scVelo, and can be partly better probably thanks to the spliced RNA-focused design.

**Inferring directed trajectories in additional featured datasets**
To further demonstrate the wide applicability of UniTVelo, we applied it to four additional datasets, and compared it to scVelo (both stochastic and dynamical models); see quantitative metrics in Table 1, Supplementary Table S3 and Supplementary Fig. S7. First, in the scNT-seq data, both scVelo and UniTVelo identified the right direction along

the stimulation time when using BRIE2 detected differential momentum genes (Supplementary Fig. S7a). However, only UniTVelo can preserve this expected direction robustly when using highly variable genes selected by scVelo (Supplementary Fig. S7b). Second, in the hindbrain (pons) of adolescent (P20) mice, scVelo suffers to find the full differentiation trajectory, while UniTVelo managed to identify a continuous path from oligodendrocyte precursors (OPCs) to committed oligodendrocyte precursors (COPs) followed by newly formed oligodendrocytes (NFOLs) and myelin-forming oligodendrocytes (MFOLs, Supplementary Fig. S7c).

Lastly, on the two pancreas datasets, both UniTVelo (the independent mode) and scVelo identify the major trajectories, with minor differences on the cycling progenitors part and terminal areas (Supplementary Fig. S7d,e). Interestingly, UniTVelo captures a subtle transition pattern from a subset of Fev+ cells (pre-endocrine, orange cluster) to Delta cells (dark purple cluster) on both datasets, which is consistent with the discovery reported by using CellRank that is tailored for cell fate mapping[28].

## Discussion
We present UniTVelo in this work which offers an alternative approach to quantify the RNA velocity of each gene and cell. This method enjoys a top-down design with an RBF to focus on the dynamics of the spliced RNAs, which are the major contents in most scRNA-seq data. This design relaxes the dynamics of transcription rates from a commonly used step function to more flexible profiles. It aligns with a recently proposed framework MultiVelo to use chromatin accessibility to predict the transcription rate[29]. Also, this top-down design brings convenience to model spliced RNAs with a broader family of dynamic functions, including deep neural

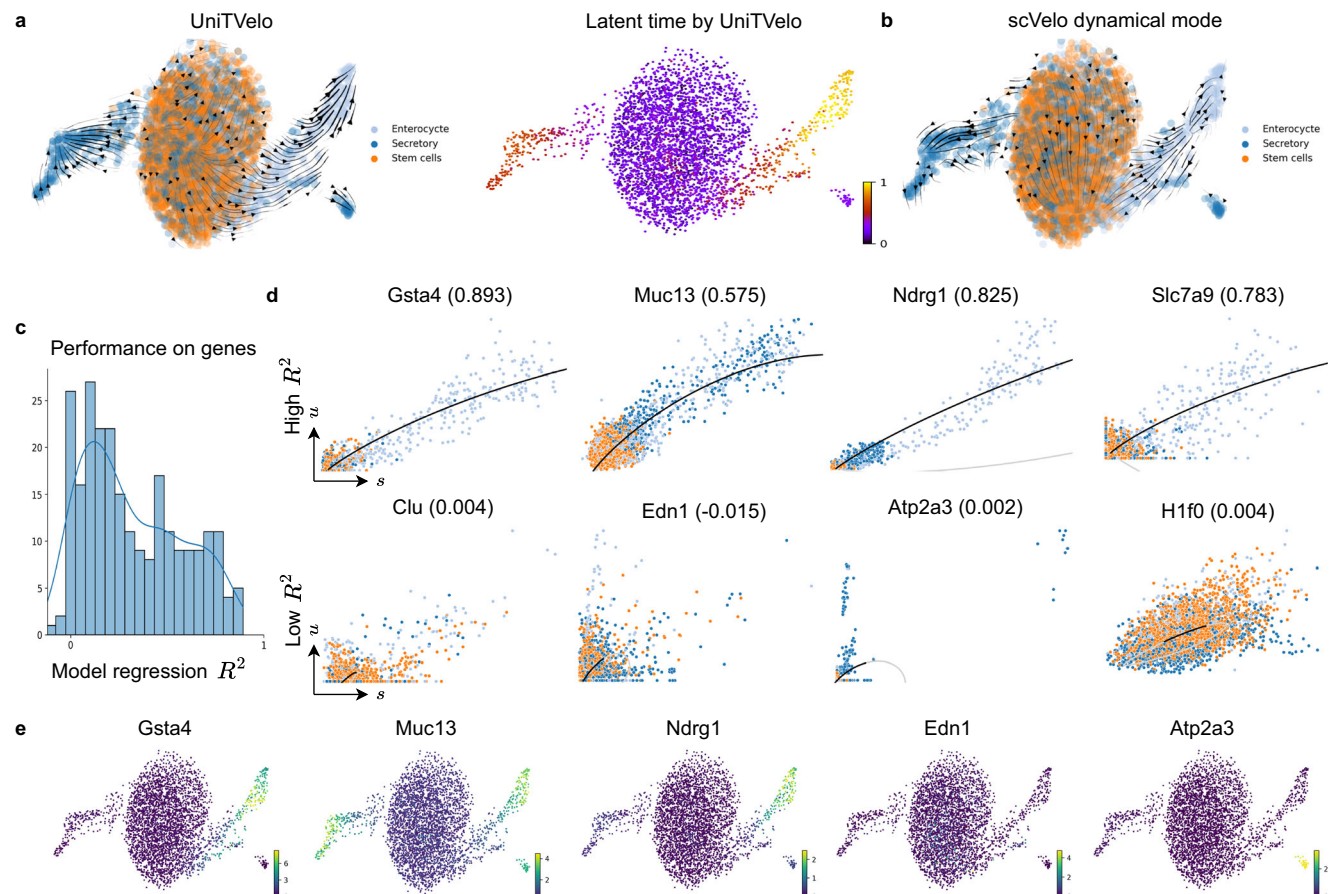

**Fig. 4 | Delineating cell fate transitions in intestinal organoid. a** Intestinal organoid differentiation dataset is acquired with scEU-seq. Without metabolic labeling knowledge, UniTVelo clearly reveals two differentiation branches to secretory and enterocyte lineage from stem cells in both velocity field (left) and inferred latent time (right). **b** On the contrary, scVelo reveals a distorted or reversed directionality under same data inputs. **c** Histogram of the distribution of model regression $R^2$ from UniTVelo on each gene of dataset. **d** Example genes with higher $R^2$ and lower $R^2$ are shown respectively. Better-fitted genes tend to have a more evident expression trend in certain lineage, which could presumably be used to interpret the inferred trajectory. Poorly fitted genes tend to exhibit less obvious traits or less abundant. Colors for each cell type are in accordance with **a**. **e** Examples of expression profiles on genes with high $R^2$ or low $R^2$.

networks that have high complexity, and to predict the velocity even entirely without unspliced RNAs[25].

Additionally, UniTVelo supports a unified time when estimating the expression dynamics for the whole transcriptome. Critically, this setting allows incorporating dynamic genes even with weak kinetic

**Table 1 | Performance comparison across datasets between UniTVelo and scVelo**

| Datasets | scVelo (Sto) | scVelo (Dyn) | UniTVelo |
|---|---|---|---|
| Pancreas (with cell cycle) | **0.516** | 0.462 | 0.497 |
| Pancreas (without cell cycle) | 0.501 | 0.497 | **0.523** |
| Dentate Gyrus | −0.855 | 0.158 | **0.746** |
| Retina development | **0.657** | 0.467 | 0.638 |
| scNT-seq | 0.242 | 0.183 | **0.483** |
| Intestinal organoid | 0.048 | 0.065 | **0.594** |
| Hindbrain (pons) | 0.347 | 0.074 | **0.609** |
| Mouse erythroid | 0.129 | −0.445 | **0.793** |
| Human erythroid | −0.332 | −0.261 | **0.465** |
| Human bone marrow | −0.817 | −0.839 | **0.804** |

Cross-boundary Direction Correctness (CBDir) is used to evaluate the transition correctness given groundtruth, see Methods. The best performance in each metric is highlighted in bold font. Sto: scVelo's stochastic mode. Dyn scVelo's dynamical mode.

information. Thanks to these settings, UniTVelo substantially increases the applicability of current RNA velocity methods, particularly in erythroid maturation, hematopoiesis, intestinal organoid, and other technically challenging datasets.

Besides the unified-time mode, UniTVelo also supports a gene-independent mode to assign the latent time to each gene independently, similar to scVelo. In general, the unified mode allows aggregating information for all genes, hence reinforcing the directionality in the trajectory inference, e.g., in intestinal organoid differentiation. On the other hand, the independent mode can also be beneficial thanks to its higher flexibility despite the risk of over-fitting on some genes. As demonstrated above, for datasets with a high signal-to-noise ratio, the independent mode remains capable of detecting complex differentiation with cycling proliferation, multiple branches, and sparse populations. In future, we anticipate a coherent framework to combine these two modes, e.g., by learning multiple-dimensional time variables that are related to the low-dimensional representation of gene space in VeloAE[17].

Although we have enhanced the quantification of RNA velocity, there are still multiple technical challenges to address. First, it remains an open challenge to select dynamically informed genes for projecting cell transitions from the velocity vector, particularly in an unsupervised manner. Here, we illustrated that the coefficient of determination $R^2$ could be an informative indicator of goodness-of-fit and further reveal the functions of a certain differentiation path. On the other hand, we observed a set of genes that have strong dynamical

patterns on spliced RNAs but limited deviation from the balance of unspliced and spliced RNAs. We termed these genes as stably and monotonically changed genes. Given their contributions to the unified mode in UniTVelo, we expect more systematic methods for identifying and integrating them to further strengthen the RNA velocity analysis, which shares a similar philosophy of CellRank by combining transcriptome similarity and velocity-based transition[28]. We also identified a few genes with reversed transient (U-shaped) behavior (Supplementary Fig. S8), although they only account for a small subset of total genes. In case such reverse transient genes are dominant, one may consider relaxing the sign for RBF to expand its modeling capacity.

Second, a few pre-processing steps still need to be evaluated or enhanced, for example, the counting of spliced and unspliced RNAs. Currently, the counting methods of unspliced reads still remain in a high discrepancy between each other[30], and the possible over-counting may explain that a few genes dramatically affect cell transition metrics from the RNA velocity vector.

Third, Bayesian methods have been shown as an principled way for modeling the high variability in single-cell data[31]. Considering that the latent time plays a key role in the quantification of RNA velocity, accounting for their uncertainty in a Bayesian manner may further propagate the uncertainty in each dimension to the inferred trajectory, hence allowing us to estimate the confidence of the detected directionality.

Lastly, our current implementation benefits from GPU acceleration while its limited memory size may be a bottleneck for datasets, e.g., larger than 50,000 cells for a GPU with 12GB memory (see benchmarking in Supplementary Tables S4 and S5). In such scenarios, we introduced a sub-sampling strategy (Methods) to fit the model on a subset of cells and predict the rest of the data, which can preserve the accuracy in the densely distributed cell populations (Supplementary Fig. S9), while it remains to be further evaluated how the sub-sampling may affect the accuracy across diverse datasets. Nonetheless, one can use also CPU to run very large datasets to avoid memory bottleneck by using more running time (e.g., around 8x more time on the human erythroid dataset).

Overall, we expect that the development of RNA velocity methods will keep blooming, including versatile and user-friendly utilities like scVelo and downstream functional analysis such as CellRank.

## Methods
### Data pre-processing
The pre-processing module of all datasets analyzed in this paper follows the standard procedure of scVelo. The sequenced matrices of unspliced and spliced counts were size normalized across all cells (script below). High-quality genes were selected with a threshold that at least 20 cells have both un/spliced mRNA counts expression. Based on the principal component analysis (30 components by default), Euclidean distances were used to construct the K nearest-neighbor graph (30 neighbors by default) on logarithmized spliced mRNA counts. Due to high noise in scRNA-seq protocols, raw counts need to be smoothed before velocity estimation for variance stabilization, first-order moments were computed for each cell based on KNN graph, namely both spliced and unspliced RNA values of each cell were replaced by the average of its all neighbors. These pre-processing steps were done by scVelo (as scv) with following scripts, *scv.pp.filter_and_normalize()* and *scv.pp.moments()*.

### Genes used to calculate RNA velocity
scRNA-seq has the possibility of measuring thousands of genes simultaneously whereas limitations arise, including bias of transcript coverage, high technical noise and low capture efficiency[32]. Therefore using all expressed genes for downstream analysis is not recommended and highly variable genes (HVGs) which contribute to cell-cell variation were selected (default 2,000).

We further selected informative genes with similar settings in scVelo (otherwise specified). The feature space is further filtered by choosing genes with a positive coefficient ($\gamma > 0.01$) between spliced and unspliced counts and a positive coefficient of determination ($R^2 > 0.01$) as well. Given the low capture rate in unspliced mRNA counts, certain genes might exhibit irregular high discrepancy between un/spliced expression profiles, which need to be further examined. Therefore, velocity genes with extreme ratio of standard deviations between unspliced versus spliced RNAs were filtered out ($\sigma_{ratio} < 0.03$) or ($\sigma_{ratio} > 3$).

However, such stringent gene filtering process may remove some genes of interest and thus limits the downstream analysis. We further introduced an optional way to expand the velocity genes during the optimization process. Specifically, we fitted a regression analysis between interim inferred cell time and spliced mRNA reads of each gene, and genes with a $R^2$ higher than the user-defined threshold (*config.AGENES_R2*) will be added to the subsequent model calculations. This allows post-analysis on more genes and the RNA velocity of those genes can be inferred as well.

### Modeling transcriptional dynamics and RNA velocity
Different from the existing RNA velocity methods, here we formulated a spliced RNAs oriented computational framework that maps the variables in a top-down manner. Specifically, we first defined the spliced RNAs via a time function $s_g(t) = f(t; \theta_g)$. Then a linear dynamical system is adopted to derive the expectation of unspliced RNAs. The main requirement of $f(t; \theta)$ is second-order differentiable, and in general it can be broadly chosen, hence sigmoid, bell-shape functions or flexible neural networks are well suitable to describe the gene expression behavior in time. Here, by using a radial basis function (RBF) as $f(t)$ by default, we can write the model explicitly as follows,

$$\begin{aligned} s_g(t) &= h_g * e^{-a_g * (t_{ng} - \tau_g)^2} + o_g \\ u_g(t) &= \frac{s'_g(t) + \gamma_g * s_g(t)}{\beta_g} + i_g \end{aligned} \quad (2)$$

with gene-specific parameters ($h_g, a_g, \tau_g, o_g$) to form a RBF based expression model, linking parameters ($\gamma_g, \beta_g, i_g$) to bond un/spliced data and cell-specific time points $t_{ng} \in (0, 1)$. Here, $u_g(t)$ and $s_g(t)$ is denoted as the mean function of un/spliced counts (i.e., the predicted expression value) along time, which indicate how expression level of genes change along differentiation time whilst $u_g(t)$ is a simple transformation from Eq.(1).

Although gene's activity and its accompanying transcription regulation in a biological system is sophisticated, theoretically they have to go through induction phase followed by repression phase. The vital difference between individual genes within a biological process is the activation time and peak time. To utilize this trait, we purpose Gaussian-like mean kernel function as stated in Eq.(2) with ($h_g, a_g, \tau_g, o_g$) to describe the expression strength, scaling factor of kernel to control the activation time, peak time of that particular gene and any offsets it may contain. Once we obtain the mean function, the RNA velocity can be directly calculated via the first derivative of spliced mRNA counts,

$$velocity = \frac{ds_g(t)}{dt} = s_g(t) * (-2a_g * (t_{ng} - \tau_g)). \quad (3)$$

Of note, this calculation of RNA velocity only relies on the fitted function of spliced RNAs. It is equivalent to the definition in scVelo or velocyto but replaced the observed unspliced counts to its predicted value, which may be able to mitigate the high stochasticity in the measures of unspliced counts.

For the assignment of time points, they were by default confined and rescaled within the range from 0 to 1, to simulate the progress

from immature to mature. Unified-time mode and independent mode, in general, share the same model structure and optimization procedure, though unified-time mode implemented a stringent gene-shared time setting to account for genes with stable and monotonic changes.

## Parameter inference of UniTVelo

To unveil the subtle splicing kinetics, two sets of parameters need to be inferred, the gene-specific $\theta_g$ that defines the shape of mean function of both un/spliced reads and the cell-specific time points $t_{ng}$ which are ordered sequentially and assigned by projecting the observation to regression line. Taken together, behavior of each gene along the differentiation trajectory could be recovered and RNA velocity could be inferred as well.

Let $u_i$, $s_i$ with $i \in (1, \ldots, N)$ be the normalized observation of un/spliced counts for a particular gene. Similarly, let $\hat{x}(t) = (\hat{u}(t), \hat{s}(t))$ be the model's estimation of un/spliced counts of that gene. The aim of this inference model is to find a certain set of parameters in which its related mean function trajectory could describe observations the best, essentially a non-linear regression. For the loss function that connects predicted mean function with observation data, we use signed Euclidean distance $e_i = (e_{ui}, e_{si})$ as residuals under the assumptions that the residuals are normally distributed with $e_i \sim N(0, \sigma^2)$. We assume the gene-specific $\sigma_g = (\sigma_{ug}, \sigma_{sg})$ across cells are distinct between un/spliced mean functions to account for stochasticity. The combined likelihood function of both unspliced and spliced mRNA counts for a particular gene can be derived in the following equation,

$$\mathcal{L}(\theta_g) = b_g * \exp\left(-\pi b_g^2 \sum_i^N |x_i^{obs} - \hat{x}_i(t)|^2\right), \tag{4}$$

where

$$b_g = \frac{1}{\sqrt{2\pi}\sigma_g}$$
$$\theta_g = (h_g, a_g, \tau_g, \gamma_g, \beta_g, b_g, o_g, i_g) \tag{5}$$

Subsequently, we need to iteratively minimize the negative log-likelihood to find the optimal parameter settings of phase portrait which is given by

$$l(\theta_g) = \pi b_g^2 \sum_i^N |x_i^{obs} - \hat{x}_i(t)|^2 - log(b_g). \tag{6}$$

The initialization of model parameters could either be manually defined or inferred from observations. Here we used a non-informative initial value with $t_g = 0.5$, which assumes all genes have experienced induction and repression during differentiation.

The objective function is optimized by Gradient Descent algorithm after meaningful parameters are initialized. The algorithm is applied with Adam optimizer which basically contains the following two steps,

- Given $\hat{x}_i(\theta_g | t)$ parametrized by the assigned gene-specific time points $t_{ng}$ for each cell, the optimizer computes gradients of objective function and updates gene-related parameters $\theta_g$ iteratively. This procedure occurs for the majority of total iterations.
- Periodically, the algorithm hold $\theta_g$ fixed and re-assign $t_{ng}$ by minimizing the Eculidean distance between observed $x_i^{obs}$ and updated phase trajectories, using grid search.

Algorithm will terminate if all parameters reach the predefined convergence criteria (proportional change on loss < $10^{-4}$) or the maximum number of iterations. The reference running time (using one GPU card, NVIDIA GeForce RTX 2080 Ti) and memory usage for each

dataset compared with scVelo dynamical mode can be found via Supplementary Table S4 and S5.

## Gene-shared cell time in unified-time mode

Allocated time points of each cell were assigned and updated with fixed iteration intervals. This predictive algorithm with RBF linking function allows us to order cells ranging from 0 (progenitor cells) to 1 (differentiated cells) and genes along the trajectory. Though both modes shared the same model structure, they differ in whether the gene-specific time is aggregated into a single gene-shared time points.

The unified-time mode was firstly introduced with gene-shared time points and initially designed to applied on datasets in which genes have less kinetic information. We discovered in most cell lineages, a large proportion of genes rarely show the dynamical characteristic described in ref. 12, for instance linear expression patterns with monotonous rise or monotonous decline. This pattern motivates us to question whether gene-specific time, as used in scVelo and our independent mode, is too flexible and using steady-state regression line to separate cells assigned to induction and repression phases suffers from weak directionality.

To address the above issues, the unified-time mode adopted the full dynamic parameters $\theta_g$ during optimization which has less constraints on the phase portraits of unspliced and spliced counts. Consequently, parameter $\tau_g$ vividly reflects genes' behavior during differentiation. In detail, $\tau_g$ should have three configurations of gene's activity: repressed ($\tau_g \leq 0$), induction ($\tau_g \geq 1$) or a combination of induction and repression (transient, $0 < \tau_g < 1$).

For each gene, the time assignment of each cell was not directly based on projection to phase portraits, instead cells were re-ordered by relevant positions after projection, to make correct alignment between genes. After cell re-ordering by their relative positions, a gene-shared time point for each individual cell was calculated by

$$t_n = \frac{1}{G} * \sum_g^G \mathbb{Q}[t_{ng}], \tag{7}$$

where $\mathbb{Q}[\cdot]$ denotes the quantile of $t_{ng}$ for each gene $g$. Additionally, this procedure also supports a denoise by projecting the gene space to a lower dimensional space (e.g., 50) by singular vector composition, before averaging across all dimensions.

## Gene-specific cell time in independent mode

Whilst unified-time mode has proven its ability through multiple scenarios, the performance was impaired in some other biological systems, like sparse cell types and cell proliferation cycle to elaborate. We hypothesized that the former setting emphasis more on those genes with stable and monotonic changes by aggregating cell time and might neglect the fact that some genes exhibit strong or complex dynamical pattern.

Therefore for independent mode, we made the following alternations on parameters to be optimized,

- For gene-specific parameters, $(\tau_g, o_g, i_g)$ were fixed at $(0.5, 0, 0)$, giving more constraints on phase portraits and this indirectly fixed the starting point of phase portraits, similar as scVelo. Compare with the flexible portraits in unified-time mode which are more likely to capture non-kinetic genes, this assumes genes to exhibit a clear induction and repression process during model fitting.
- For cell-specific time, we adopted the similar time assignment procedure as in the unified-time mode, to project cells onto phase portraits by minimizing the Euclidean distance. However, instead of pooling the time together to form cell-specific time vector, we directly used gene-specific time matrix during model

optimization. Since each gene was independent, cell re-ordering was not needed.

## Choose the suitable modes

The choice of the mode is a hyper-parameter that user can set manually. For the majority of biological systems tested in this paper, Uni-TVelo by default used unified-time mode, otherwise specified. We provided a utility function (*utils.choose_mode*) for identifying complicated datasets and suggesting the mode to use. The complicated datasets are defined with the following criteria,

- Datasets with cell cycle phase included (e.g., Supplementary Fig. S6b, S7d). Determining whether a dataset contains cell cycle stage might not be straightforward, one potential way is to check number of cycle genes[33] which are highly variable (Supplementary Table S2). Generally, we observed that for cycle related datasets, number of cycle genes in both S and G2M phases which are highly variable are significantly higher than other datasets. This could be a potential way to identify cycle-related datasets and help users to choose modes. Specifically, we evaluate cycle genes after selecting the HVGs, follows the procedure as scVelo[12] and Seurat[34] do. And datasets with number of cycle genes in either S or G2M phases higher than half (an adjustable hyper-parameter) of the gene lists defined in ref. 33 are considered with cell cycle included.
- Datasets with sparse cell types included (e.g., Supplementary Fig. S6a). Normally scRNA-seq data and the related velocity streamlines are visualized on embeddings like UMAP, which reflects the similarity of expression profiles of various cell types. To elaborate, sparsity refers to a few clusters scattered around with no obvious connections with others, meaning the proportion of its neighbor cells belong to the same cluster should be quite high. For now, we define if there are more than 2 clusters with more than 95% of its neighbor cells are within the same cluster (both are adjustable hyper-parameter), we consider using independent mode.

## Down-sampling and prediction strategy

In scenarios when hardware is a bottleneck for large-scale datasets (e.g., limited GPU resources), we also provide a utility script to down-sample the original dataset, run the model on down-sampled data and predict RNA velocity and cellular time for the rest of cells. This down-sampling strategy provided by UniTVelo considers the problem of rare cell populations in scRNA-seq datasets, by providing a user-defined threshold parameter specifying the minimal number of cells (e.g., 50) within each cluster to keep, and a parameter representing the percentage of sampling.

After generating relevant gene-specific parameters using down-sampled data, the model would then predict RNA velocity and cell time by the same projection process used in normal training process (the second part of the parameter inference step). In Supplementary Fig. S9 we have proved this strategy is robust to dense datasets and could achieve a satisfactorily accuracy compared with full batch.

## Model evaluation metrics

Identifying informative genes which explain the inferred cell trajectory, we employed the coefficient of determinant $R^2$, that are commonly used in the regression realm, to indicate the goodness-of-fit. Here, it only focuses on the spliced RNAs, so $R^2$ denotes the proportion of variance explained by the time function $f(t)$. Generally speaking, a high $R^2$ means the time function well captures the dynamical pattern of a certain gene, hence reflecting the estimated biological progression.

Furthermore, to assess the model performance with expected trajectory directions, two quantitative evaluation metrics are used in this paper to compare between different algorithms as proposed in

ref. 17: cross-boundary direction correctness (CBDir) and in-cluster coherence (ICCoh). To elaborate, CBDir measures the correctness of transitions from a source cluster to target cluster using boundary cells given ground truth. Here boundary of source cluster refers to cells in that cluster that are neighbors of target cluster and vice versa. Boundary cells are used because they reflects the biological development in a short period of time and CBDir is calculated via,

$$CBDir(c) = \frac{1}{Norm} \sum_{c' \in C_A \cap N(c)} \frac{\mathbf{v}_c \cdot (\mathbf{x}_{c'} - \mathbf{x}_c)}{|\mathbf{v}_c| \cdot |\mathbf{x}_{c'} - \mathbf{x}_c|}$$
$$Norm = |\{c' \in C_A \cap N(c)\}| \qquad (8)$$

where $C_A$ is sets of cells in target cluster A, $N(c)$ stands for the neighboring cells of specified cell $c$. $\mathbf{v}_c, \mathbf{x}_{c'}, \mathbf{x}_c$ are the low-dimensional vectors representing computed velocity and positions of cell $c$ and $c'$. Therefore, $\mathbf{x}_{c'} - \mathbf{x}_c$ is the displacement is space during the short period of time. This metric requires the ground truth directions to be input and thus could reflect the reliability of RNA velocity model. ICCoh, on the other hand, is calculated using cosine similarity scoring among cells within the homogeneous cluster. This shows the smoothness of velocity within cluster and therefore, could achieve a high score even if the directions are reversed,

$$ICCoh(c) = \frac{1}{Norm} \sum_{c' \in C_A \cap N(c)} \frac{\mathbf{v}_c \cdot \mathbf{v}_{c'}}{|\mathbf{v}_c| \cdot |\mathbf{v}_{c'}|} \qquad (9)$$

This two quantitative metrics should be used in combination, velocities of both cross-cluster correctness and in-cluster consistency can be calculated, which enables the performance comparison between different RNA velocity algorithms.

## Reporting summary

Further information on research design is available in the Nature Research Reporting Summary linked to this article.

## Data availability

Mouse and human erythroid differentiation: Erythroid lineage which derived from human and mouse gastrulation process. Cells are sequenced using 10X Genomics V2 sequencing protocol based on droplet method. Human erythroid lineage is available from[15] whilst mouse gastrulation subset is incorporated by *scv.datasets.gastrulation_erythroid()*. Human bone marrow hematopoieses: The raw data counts as well as associated experimental details can be accessed through the Human Cell Atlas data portal under Human Hematopoietic Profiling project. Processed data is integrated in scVelo via *scv.datasets.bonemarrow()*. Intestinal organoid differentiation: Datasets have been deposited in GEO with accession number GSE128365. Data and labels used for RNA velocity analysis is available upon requesting[18,26]. Dentate gyrus neurogenesis development: Experiment of Dentate Gyrus development comprises two time points (P12 and P35) using droplet-based scRNA-seq protocol. Details can be accessed by *scv.datasets.dentategyrus()*. Mouse developing retina: Raw data of mouse developing retina was sequenced by 10x Chromium and has been deposited in GEO under GSM3466902. Processed data for velocity analysis and result reproduction is downwded from Kharchenko Lab. Neuron genesis with KCI stimulation: Sequenced data within this study is available in Gene Expression Omnibus (GEO) under accession number GSE141851. Processed counts and annotations is available upon requesting[13]. Hindbrain (pons) of adolescent mice: The differentiation of oligodendrocyte lineage and its associated myelination process is demonstrated. Counts metrics and annotations in .rds format are acquired from Kharchenko Lab. Pancreatic endocrinogenesis: Pancreatic epithelial cells were sampled at day 15.5 from embryonic with four possible terminal states. Processed data is acquired from *scv.datasets.pancreas()*.

## Code availability

UniTVelo is freely available as Python package at https://github.com/StatBiomed/UniTVelo and https://doi.org/10.5281/ZENODO.7112387 with both unified-time mode and independent mode implemented. Detailed workflows to reproduce figures and results in this paper are written as Jupyter notebook in the repository.

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

## Acknowledgements

We thank Melania Barile for the fruitful discussion. We acknowledge support from the University of Hong Kong and its Li Ka Shing Faculty of Medicine through a startup fund and a seed fund.

## Author contributions

Y.H. conceived the ideas and designed the study. M.G. developed the algorithm, implemented UniTVelo, and analyzed the data. C.Q. contributed to troubleshooting. M.G. and Y.H. wrote the manuscript. All authors read and approved the final manuscript.

## Competing interests

The authors declare no competing interests.

## Additional information

**Correspondence and requests** for materials should be addressed to Yuanhua Huang.

