## [Peer Review File · Nature Communications]

REVIEWER COMMENTS

Reviewer #1 (Remarks to the Author):

RNA velocity is a useful tool for scRNAseq but it can be inaccurate or inconsistent due to signal-to-noise ratio in unspliced mRNAs. To overcome this, the authors presented a new way of obtaining RNA velocity by using the dynamic changes of spliced reads instead of directly using unspliced reads. Spliced mRNAs were modeled using RBF. The cell time is assigned in the space drawn by unspliced versus spliced reads. The authors evaluate the performance using various datasets including erythroid maturation.

For the erythroid dataset, the authors claimed that UniTVelo overcomes the limitation of scVelo potentially due to transcriptional boosting in the later stage of the development.

For the bone marrow development datasets, UniTVelo showed velocity towards the three terminal stages while scVelo showed some reversed direction. UniTVelo found repressive dynamics of Cd44, Celf2 and Taok3. UniTVelo also showed the best performance in the quantification test using CDir and ICCoh. The manuscript is well written and the hypothesis is well reasoned. This tool will be useful for scientific community.

minor.

Please spell out MEMP. Please provide the references.

RBF can model transient status (up and down over time). Can it model the transient status of down and up? If it does not matter, how the direction of the velocity can be handled.

Reviewer #2 (Remarks to the Author):

RNA velocity that captures the short-term gene expression change (based on spliced and unspliced RNAs) provides new opportunities to reconstruct the cellular trajectories (other than the pseudo-time

trajectory inference based on intercellular expression difference). RNA velocity or scVelo package were commonly used to infer the RNA velocity vectors that could be fed into other downstream analyses and showed great success in the past few years. In this work, the authors proposed a significant extension of the scVelo package-uniTVelo, which has demonstrated superior performance in multiple single-cell datasets with ground-truth known.

Two major innovations of the uniTVelo that the authors claimed are: (1) it's a top-down design. In other words, the authors first fit the expression profiles of spliced RNAs with a radial basis function. With the learned expression profiles, the authors then derive the dynamics of unspliced RNAs and transcription rates via the first-order dynamic differential equations. This differs from the bottom-down strategy employed by the original scVelo method, which first defines/learns a transcription rate and then derives the expression profile with the differential equations.

(2) The second innovation is the introduction of a unified latent time for a cell. Unlike scVelo, uniTVelo, as suggested in the name, aggregates the dynamic information across all genes to infer the temporal ordering of cells, which allows the effective incorporation of stably and monotonically changed genes.

The method described in this manuscript is clear and logically sound. It is also easy to follow the method. However, I still have a few significant concerns regarding the proposed method that I hope the authors can address in the revised version.

(1) For the 1st innovation/advantage compared to scVelo framework, the authors claimed that their top-down strategy is more flexible and could enable a better representation of complex gene expression profiles. Via comparison in multiple real single-cell datasets, the authors did demonstrate that their method is able to present the velocity that is more consistent with known knowledge. However, there is no direct comparison between the expression profile "fitness" of the top-down or bottom-up strategies. In those datasets where uniTVelo has superior performance, can authors show indeed that the dynamic gene expression profiles could be better modeled with the top-down strategy (compared with the scVelo bottom-up strategy)? I understand that the authors did provide some examples in Figures 2-4. I am wondering whether the authors could provide a systematic comparison (e.g., of the entire transcriptome), which could better support the first innovation that the authors claimed.

(2) As the second major innovation, uniTVelo learns a unified latent time across the whole transcriptome for the cell, which allows to effectively incorporate stably and monotonically changed genes. I understand that the author employs this strategy to avoid potential over-fitting. However, this actually reduces the flexibility of the model and thus could hamper the performance in complex single-cell datasets. The authors did realize this potential drawback, and they proposed two separate models for different tasks. For most applications, the standard unified time model should be used, while the independent model should be used for more complicated datasets. This indeed solves the problem, at least partially. However, it would be difficult for the users to decide which model should be employed

for their specific datasets. The authors provided some guidance "independent model of UniTVelo is intended for more complicated datasets, for instance, datasets with cell cycle or sparse cell types included", but the guidance is quite vague. Therefore, more clear guidance should be provided to instruct the users on when to use the independent model (preferably a utility script to help choose the mode). This should greatly boost the usefulness of the proposed UniTVelo method.

(3) Another important aspect that should be covered in the manuscript is the running time efficiency. UniTVelo, as a scVelo extension, utilizes an EM strategy to learn the parameters for the model. Therefore, I would assume that the running time (and memory cost) would be significantly higher compared to the original scVelo. This may not be critical for small or regular size single-cell RNA-seq data (e.g., 3-30k cells). Nevertheless, this could become very important for the application of big datasets (e.g., for datasets with 300k-1M cells). A running time (memory) benchmarking would be needed to address this concern.

A few minor comments:

(1) In the preprocessing, the authors selected 2000 most variable genes, which were further filtered with other criteria. This is a common strategy employed by many single-cell methods, including the original scVelo. However, such stringent gene filtering often removes many important genes of interest and thus limits the downstream analysis. While I agree with the authors that inferring RNA velocity based on those most informative genes would be a good strategy, allowing post-analysis on the other genes (based on the inferred RNA velocity) may be very helpful for the downstream biological examinations. The authors could consider adding such functions to improve the practical usage of their method.

(2) The evaluation metrics (CBDir) and ICCoh should be briefly described, besides the citation of the manuscript that the authors published last year.

Point-by-point response

Reviewer 1: pages 1-2

Reviewer 2: pages 3-7

Reviewer #1 (Remarks to the Author):

RNA velocity is a useful tool for scRNAseq but it can be inaccurate or inconsistent due to the signal-to-noise ratio in unspliced mRNAs. To overcome this, the authors presented a new way of obtaining RNA velocity by using the dynamic changes of spliced reads instead of directly using unspliced reads. Spliced mRNAs were modeled using RBF. The cell time is assigned in the space drawn by unspliced versus spliced reads. The authors evaluate the performance using various datasets including erythroid maturation.

For the erythroid dataset, the authors claimed that UniTVelo overcomes the limitation of scVelo potentially due to transcriptional boosting in the later stage of the development.

For the bone marrow development datasets, UniTVelo showed velocity towards the three terminal stages while scVelo showed some reversed direction. UniTVelo found repressive dynamics of Cd44, Celf2, and Taok3. UniTVelo also showed the best performance in the quantification test using CbDir and ICCoh. The manuscript is well written, and the hypothesis is well reasoned. This tool will be useful for the scientific community.

Response: we thank the reviewer for the appreciation of our work.

(1) Please spell out MEMP. Please provide the references.

Response: Added in the revised manuscript (p.2); thanks.

(2) RBF can model transient status (up and down over time). Can it model the transient status of down and up? If it does not matter, how the direction of the velocity can be handled.

Response: Thank you for this fundamental question.

By default, RBF indeed could only model induction phase (monotonic increasing), repression phase (monotonic decreasing) and transient phase (up and down over time), and it is difficult to model the transient status of down and up over time for the time being, given the convex nature of the shape.

To test the hypothesis that whether the abovementioned genes exist and to what extent those genes affect the overall directionality of the velocity field, we have used quadratic function to model the expression profiles of each gene along with inferred cell time and compared the performance with standard RBF function (without unspliced reads). If the transient status of down and up exists, then the parabola should open upwards (positive coefficient) and have a R^2 at least slightly higher than that of RBF model.

We've tested this on available datasets and came to a conclusion that generally, genes with reversed transient status are quite rare compared to the total number of genes used to construct the velocity graph, normally less than 10%, except for scEU-seq and one of

the Pancreas data (but with quite a few false positives exist due to unbalanced number of cells in each cluster). Details are shown with the following scatter plots,

Here shows a comprehensive comparison for each gene. The numbers behind title are the ratio of identified reversed genes divided by the total number of genes. Note that the threshold to classify whether a gene is reversed transient or not is low (R^2 of quadratic - R^2 of RBF > 0.075) and can result in false positives, but still the ratio is relatively small. We anticipate a smaller ratio if a more stringent threshold is used. Below are 3 example genes from scEU Organoids dataset which are identified as reverse transient whilst they are more likely in an induction state (colors represent different cell clusters).

To conclude, in our current implementation, we keep a positive sign for RBF, hence cannot model the reverse transient shape. We think this setting is beneficial, considering that very limited genes have strong reverse transient dynamics. In case there are such scenarios, we can easily relax the sign of RBF to account for the reverse transient shape. We have added this figure in Supp Fig. S8 and more discussion on p.8.

Reviewer #2 (Remarks to the Author):

RNA velocity that captures the short-term gene expression change (based on spliced and unspliced RNAs) provides new opportunities to reconstruct the cellular trajectories (other than the pseudo-time trajectory inference based on intercellular expression difference). RNA velocity or scVelo package were commonly used to infer the RNA velocity vectors that could be fed into other downstream analyses and showed great success in the past few years. In this work, the authors proposed a significant extension of the scVelo package-uniTVelo, which has demonstrated superior performance in multiple single-cell datasets with ground-truth known.

Two major innovations of the UniTVelo that the authors claimed are:

(1) it is a top-down design. In other words, the authors first fit the expression profiles of spliced RNAs with a radial basis function. With the learned expression profiles, the authors then derive the dynamics of unspliced RNAs and transcription rates via the first-order dynamic differential equations. This differs from the bottom-down strategy employed by the original scVelo method, which first defines/learns a transcription rate and then derives the expression profile with the differential equations.

(2) The second innovation is the introduction of a unified latent time for a cell. Unlike scVelo, UniTVelo, as suggested in the name, aggregates the dynamic information across all genes to infer the temporal ordering of cells, which allows the effective incorporation of stably and monotonically changed genes.

Response: thank you for the precise summary of our innovations.

The method described in this manuscript is clear and logically sound. It is also easy to follow the method. However, I still have a few significant concerns regarding the proposed method that I hope the authors can address in the revised version.

(1) For the 1st innovation/advantage compared to scVelo framework, the authors claimed that their top-down strategy is more flexible and could enable a better representation of complex gene expression profiles. Via comparison in multiple real single-cell datasets, the authors did demonstrate that their method is able to present the velocity that is more consistent with known knowledge. However, there is no direct comparison between the expression profile fitness of the top-down or bottom-up strategies. In those datasets where UniTVelo has superior performance, can authors show indeed that the dynamic gene expression profiles could be better modeled with the top-down strategy (compared with the scVelo bottom-up strategy)? I understand that the authors did provide some examples in Figures 2-4. I am wondering whether the authors could provide a systematic comparison (e.g., of the entire transcriptome), which could better support the first innovation that the authors claimed.

Response: Thank you for the very good questions and suggestions.

UniTVelo has two separate modes, of which the unified-time mode periodically revises the time matrix during the optimization process, resulting in the assigned time and phase portraits for each cell and gene being largely affected by an overall representation. Thus,

it might be difficult to conduct a systematic comparison of the entire transcriptome in gene-specific level.

However, the independent mode of UniTVelo, aka the top-down strategy, fits each gene individually which shares similar logic with dynamical mode of scVelo, the bottom-up method. To directly compare the expression profile fitness of these two strategies, we use diffusion pseudotime as reference (by specifying the expected root cells) and calculate the spearman correlation with the assigned gene-specific time matrix across the entire transcriptome. Results are shown in the following figure,

UniTVelo's strategy has better performance than scVelo in 6 datasets (Erythroid Mouse, Erythroid Human, Human BoneMarrow, Hindbrain, scEU Organoids and DentateGyrus), former 5 of which have been demonstrated with superior performance with unified-time mode compared with scVelo, the only exception is the scNT dataset which UniTVelo has better performance with unified-time mode whilst scVelo fits better on gene level.

Although the difference of overall performance for 10 datasets shown is subtle between UniTVelo and scVelo, the comparison is done within the independent mode, for those datasets ought to be run with unified-time mode, we anticipate a much higher spearman correlation (Erythroid Mouse 0.986, Erythroid Human 0.978, Human BoneMarrow 0.811, Hindbrain pons 0.868, scEU Organoids 0.486, scNT Neuron 0.580).

Overall, we hope the reviewer agrees with us that our top-down strategy has a comparable performance to the bottom-up strategy, if not better, in a side-by-side comparison. On the other hand, our top-down strategy introduces computational convenience not only to be compatible with the unified-time framework but also to support broad families of dynamical functions though we only demonstrate RBF here. We have added this figure in Supp. Fig S2 and more discussion on p.2 in this revision.

(2) As the second major innovation, UniTVelo learns a unified latent time across the whole transcriptome for the cell, which allows for effectively incorporating stably and monotonically changed genes. I understand that the author employs this strategy to avoid

potential over-fitting. However, this actually reduces the flexibility of the model and thus could hamper the performance in complex single-cell datasets. The authors did realize this potential drawback, and they proposed two separate models for different tasks. For most applications, the standard unified time model should be used, while the independent model should be used for more complicated datasets. This indeed solves the problem, at least partially. However, it would be difficult for the users to decide which model should be employed for their specific datasets. The authors provided some guidance "independent model of UniTVelo is intended for more complicated datasets, for instance, datasets with cell cycle or sparse cell types included", but the guidance is quite vague. Therefore, more clear guidance should be provided to instruct the users on when to use the independent model (preferably a utility script to help choose the mode). This should greatly boost the usefulness of the proposed UniTVelo method.

Response: Exactly, that is the motivation for keeping the two modes and we agree that more detailed guidance will be helpful for users when choosing.

As mentioned, UniTVelo utilizes unified-time optimization to avoid commonly seen over-fitting as scVelo does and this has been set as default mode for most of the applications. However, for certain complicated datasets, this might hamper the model performance. Specifically, we define complicated datasets with the following criteria,

- Datasets with cell cycle included (e.g., Supplementary figures S6b and S7d). We've noticed that after basic normalization and selection of variable genes, both scVelo and Seurat have their own function (relies on a list of cycle genes defined in Irosh et al, Science, 2015.) to calculate the cell cycle scores and assign each cell with a specific cycle phase, G1, S, and G2M. However, this function would also mark cells that are not from the cell cycle, thus false positives exist, and hard to be discriminative between datasets that contain a cycle and those do not.

Determining whether a dataset contains cell cycle stage might not be straightforward, based on tested datasets, whilst we have noticed that one potential way is to check number of cycle genes which are highly variable (as shown in the following table, with a total number of 43 genes for S phase and 54 genes for G2M phase),

	S	G2M
Erythroid Mouse	15	21
Erythroid Human	0	0
Human BoneMarrow	0	0
Hindbrain (pons)	2	6
scEU Organoids	0	0
ScNT Neuron	3	2
Retina Development (with Cell Cycle)	35	42
Dentate Gyrus	4	3
Pancreas (with Cell Cycle)	22	26
Pancreas	2	3

Generally, we observed that for cell cycle related datasets, number of cycle genes in both S and G2M phases which are highly variable are significantly higher than other datasets (except for Erythroid Mouse). By setting a proper threshold, this could be a potential way to identify cycle related datasets and help users to choose which mode to use, we have prepared a small script for this purpose.

- Datasets with sparse cell types included (e.g., Supplementary figure S6a). Normally scRNA-seq data and the related velocity streamlines are visualized on embeddings like UMAP, which reflects the similarity of expression profiles of various cell types, to certain extent. Therefore, if there are quite a few clusters scattered around with no obvious connections with other clusters (identified by KNN graphs of each cell and calculate proportion of neighbor cells which are within the same cluster), we consider it as a sparse cell type dataset and recommend independent mode.

For other kind of datasets, the default mode unified time would be used. In summary, we have compiled the above specific guidance and implemented into a utility function as following script for suggesting the mode use. For less certain scenario, we also suggest user to try both. We have also added these guidances into the revised manuscript on p.10.

```
utv.utils.choose_mode(adata, label)
```

(3) Another important aspect that should be covered in the manuscript is the running time efficiency. UniTVelo, as a scVelo extension, utilizes an EM strategy to learn the parameters for the model. Therefore, I would assume that the running time (and memory cost) would be significantly higher compared to the original scVelo. This may not be critical for small or regular size single-cell RNA-seq data (e.g., 3-30k cells). Nevertheless, this could become very important for the application of big datasets (e.g., for datasets with 300k-1M cells). A running time (memory) benchmarking would be needed to address this concern.

Response: Thank you for raising this important issue.

Now, we have more clearly emphasized the benchmarking of running time and memory usage in the revised manuscript (p.9) and supplementary materials (Supp. Tables S4 and S5).

Benchmarking is conducted on regular size scRNA-seq data, the largest dataset we've tested has around 36k cells. The memory usage between two methods is subtle except UniTVelo additionally utilizes GPU for model acceleration. This approximate EM strategy requires more running time than scVelo whilst it is acceptable given the size of dataset and more accurate results.

We haven't tested UniTVelo on bigger datasets though, datasets with 300k – 1M cells. Yet, based on the available results, we anticipate a linearly increasing relationship between dataset size and running time (memory usage as well), both for UniTVelo and scVelo. There's one concern that current model is deployed on a single GPU with 12GB, we estimate that a dataset with more than 40k cells could face out of memory problem. Therefore, when there are larger datasets and the GPU memory is a bottleneck, we recommend a sub-sampling scheme and have implemented a following utility function and added discussion on p.9.

```
utv.utils.subset_adata(adata, label, proportion)
```

A few minor comments:

(1) In the preprocessing, the authors selected 2000 most variable genes, which were further filtered with other criteria. This is a common strategy employed by many single-cell

methods, including the original scVelo. However, such stringent gene filtering often removes many important genes of interest and thus limits the downstream analysis. While I agree with the authors that inferring RNA velocity based on those most informative genes would be a good strategy, allowing post-analysis on the other genes (based on the inferred RNA velocity) may be very helpful for the downstream biological examinations. The authors could consider adding such functions to improve the practical usage of their method.

Response: Thanks for the very good practical suggestion and indeed we agreed that having a utility function to re-analyze more genes can be beneficial.

Now, the stringent gene filtering process has been revised by a more flexible function in our newly pre-release version of UniTVelo model. It contains extensions at the following two levels.

First, among the 2,000 initially selected highly variable genes, we introduced a way to expand the velocity genes during the optimization process. Specifically, we fitted a linear regression between interim inferred cell time and spliced mRNA reads of each gene, and genes with a R^2 higher than the user-defined threshold (config.AGENES_R2) will be added to the subsequent model and be a part of the calculations. This allows post-analysis on more genes and the RNA velocity of those genes can be calculated as well. This approach has worked on a few datasets with dozens, or hundreds of genes being additionally identified and contributing to the overall optimization process.

Second, outside of the 2,000 variable genes, we keep most informative genes in the adata, so users can further analyze them, e.g., by correlation analysis between the spliced RNAs and the inferred latent time.

(2) The evaluation metrics (CBDir) and ICCoh should be briefly described, besides the citation of the manuscript that the authors published last year.

Response: Thanks; added in the manuscript (p.10).

REVIEWER COMMENTS

Reviewer #1 (Remarks to the Author):

All comments are well answered.

Reviewer #2 (Remarks to the Author):

The reviewers addressed most of my primary concerns. I am generally positive about the publication of the manuscript in the journal. However, there remain a few responses that I believe are insufficient.

1) For my comment on the first claimed advantage of the top-down strategy over the bottom-up strategy employed by scvelo, the authors did perform some systematic comparisons. As shown in Supplementary Figure s2, both strategies (UniTVelo vs. scVelo) present a very similar performance (almost identical mean spearman correlations with the reference diffusion pseudotime). It failed to support the authors' claim that the top-down strategy that they proposed is superior to the bottom-up strategy employed by the original scvelo. The authors also acknowledged that the performances of the two strategies are similar, and they further argued that the top-down strategy is more flexible and introduces computational convenience. However, this argument is very vague and lacks concrete evidence and supporting examples. The authors mentioned that the top-down strategy could support a broad family of dynamical functions, but it's not straightforward to see how this could benefit the inference of RNA velocity. The authors need to provide concrete evidence to back up this argument. Besides, it's also questionable to use the diffusion pseudotime as the reference as it could be quite different from the ground truth. The authors should try to find datasets with a known order of the cells (e.g., this neuro reprogramming data <https://pubmed.ncbi.nlm.nih.gov/27281220/>) and use the validated cell order as the reference to calculate the correlation. If such a dataset is difficult to find, the authors should at least compare the predictions to the reference pseudotime inferred by a few other tools (e.g., SLINGSHOT)

(2) Another big concern is the running time. As I guessed, the proposed method is very computationally expensive (requires a lot of computing resources, GPU memory, and running time). Moreover, given that a single-cell dataset with over 40k cells is very common and the size of single-cell data is ever-increasing, the scalability of the method could become a bottleneck that limits its application (~ over 5x more running time and 2x more GPU memory as shown in Table S4 and S5). The authors should provide an estimation of required memory and running time for large-scale single-cell data (say over 100k cells) so that users can know how much computing resource they will need to analyze their extensive single-

cell datasets (e.g., over 40k cells). The authors did mention a down-sampling strategy. However, this strategy will unavoidably suffer from information loss and could badly hurt the RNA velocity inference. For example, the rare cell populations could be seriously affected by the random subsampling strategy. A more sophisticated downsampling strategy might be necessary, and the authors will need to provide a systematic evaluation of different downsampling strategies to inform the users.

Minor comments:

The description of the criteria to select from two modes remains vague. I would suggest the authors add more specific guidelines. I understand that the users have implemented a utility script to choose the model. Maybe explicitly explain the concrete criteria the users had implemented. The authors also suggest running both modes. However, it would be difficult for users to judge results from which mode would be better. In addition, the running cost (time and memory) for running two modes will be prohibitive. Therefore, selecting a proper mode is not trivial and would require more explicit guidelines for users to follow.

Point-by-point response

Reviewer 1: None

Reviewer 2: Pages 1 – 6

Reviewer #1 (Remarks to the Author):

All comments are well answered.

Reviewer #2 (Remarks to the Author):

The reviewers addressed most of my primary concerns. I am generally positive about the publication of the manuscript in the journal. However, there remain a few responses that I believe are insufficient.

1) For my comment on the first claimed advantage of the top-down strategy over the bottom-up strategy employed by scvelo, the authors did perform some systematic comparisons. As shown in Supplementary Figure s2, both strategies (UniTVelo vs. scVelo) present a very similar performance (almost identical mean spearman correlations with the reference diffusion pseudotime). It failed to support the authors' claim that the top-down strategy that they proposed is superior to the bottom-up strategy employed by the scvelo. The authors also acknowledged that the performances of the two strategies are similar, and they further argued that the top-down strategy is more flexible and introduces computational convenience. However, this argument is very vague and lacks concrete evidence and supporting examples. The authors mentioned that the top-down strategy could support a broad family of dynamical functions, but it's not straightforward to see how this could benefit the inference of RNA velocity. The authors need to provide concrete evidence to back up this argument. Besides, it's also questionable to use the diffusion pseudotime as reference as it could be quite different from the ground truth. The authors should try to find datasets with a known order of cells (e.g., neuro reprogramming data <https://pubmed.ncbi.nlm.nih.gov/27281220/>) and use the validated cell order as the reference to calculate the correlation. If such a dataset is difficult to find, the authors should at least compare the predictions to the reference pseudotime inferred by a few other tools (e.g., SLINGSHOT)

Response:

Thank you for the insistence on this important challenge. Here, we further revised our manuscript in three folds.

First, we have further considered the ground truth of the differentiation time. Thanks for suggesting Slingshot, which we found that it returns similar results compared to that of diffusion pseudo-time in various types of datasets with minor exceptions, e.g., on scNT data (see the updated Supp. Fig. S2 and below). Of note, Slingshot fails to infer the reasonable trajectory / pseudo-time on scEU organoids data, so the spearman correlation is not calculated. This demonstrates the alike ability of two

algorithms in inferring gene-wise time as an individual, however, when aggregating gene-wise time to a unified-cell time, our method generally shows a better performance than that of scVelo, which are also included in the figure now.

Second, we further rephrased the texts (p.2) and added more discussions (p.7) to clarify our claim that our top-down strategy returns comparable accuracy to the conventional bottom-up design in a per-gene setting and the major benefit is its computational convenience for an easy extension to a unified mode. On the other hand, we still want to emphasize that the expression profiles between unspliced and spliced reads are complex, and it might be difficult to imitate them satisfactorily with a single approach. The top-down strategy, however, relaxes the stringent assumptions and limitations of scVelo that transcription rate is a fixed stepwise function, and the phase portraits of genes produced are not robust enough to abnormal expressions as well, e.g., transcriptional boosting or genes which are continuously in steady states.

Third, besides the convenience to support a unified mode, we still want to highlight that the top-down design enjoys higher flexibility thus we introduced more concrete literature examples to demonstrate the benefits of the top-down design. Whilst it is out of scope in this manuscript to explore how this strategy could be extended to a broader family and how it will benefit the research field explicitly, more recent literature, including scTour (Li BioRxiv 488600, 2022) could be ideal examples of this approach which utilized a similar spliced read oriented design but with deep learning architecture and facilitates RNA velocity-related research work. Therefore, we added more discussion on this point (p.7) and leave it as an open question for future evaluation and studies.

(2) Another big concern is the running time. As I guessed, the proposed method is very computationally expensive (requires a lot of computing resources, GPU memory, and running time). Moreover, given that a single-cell dataset with over 40k cells is very common and the size of single-cell data is ever-increasing, the scalability of the method could become a bottleneck that limits its application (~ over 5x more running time and 2x more GPU memory as shown in Table S4 and S5). The authors should provide an estimation of required memory and running time for large-scale single-cell data (say over 100k cells) so that users can know how much computing resource they will need to analyze their extensive single-cell datasets (e.g., over 40k cells). The authors did mention a down-sampling strategy. However, this strategy will unavoidably suffer from information loss and could badly hurt the RNA velocity inference. For example, the rare cell populations could be seriously affected by the random subsampling strategy. A more sophisticated down-sampling strategy might be

necessary, and the authors will need to provide a systematic evaluation of different down-sampling strategies to inform the users.

Response:

Thank you for further emphasising this important issue. We agreed that the running time of UniTVelo is not perfect, particularly considering the increasingly larger datasets. On the other hand, we think running for 1 hour on 36K cells is widely acceptable considering the model gives more accurate results. Given the running time is generally linear to the number of cells (Supp. Table S4), analysing 400K cells may finish within a half day which is probably not a major concern for most analyses.

Nonetheless, we still provide a down-sampling strategy in case computing resource is a bottleneck for some small labs. Here, we'd like to clarify that the down-sampling strategy provided by UniTVelo actually already considers the problem of rare cell populations in scRNA-seq datasets, by providing a user-defined threshold parameter specifying the minimal number of cells (e.g., 50) within each cluster to keep, and a parameter representing the percentage of sampling,

- 1) If the number of cells of a particular cell type is lower than threshold, all cells will be kept for model optimization.
- 2) If estimated cell number after sampling is lower than threshold (e.g., randomly select 50% of 80 cells which is 40 cells), then 50 cells will be selected.
- 3) For other clusters, the utility script will sample cells randomly based on pre-defined percentage.

In this revision, we also provide a utility script for using the down-sampled data to predict RNA velocity and cellular time for the rest of the cells, hence having all cells for downstream analysis (p.8 and Supp. Fig. S9 in p.21).

Here, we use the erythroid human dataset as an example and briefly describe the results. This dataset originally has 37k cells which use approximately **1 hour** to run the model with GPU acceleration. We first randomly sampled 10% of total cells using the down-sampling strategy mentioned above, which is around 3.6k cells. Then the model is applied to this subset, and RNA velocity and unified time are inferred (upper panel of a and b; running time around 10mins). Then we use parameters generated to predict the relevant RNA velocity and unified time for the rest of the cells (lower panels of a and b; running time 3mins). To assess the performance of this down-sampling / prediction strategy with ground truth (model fitted with full batch of data), we tried two folds of comparison,

- We have compared the inferred time for each cell between prediction and full batch, the scatter plot showed a strong correlation status.
- For the RNA velocities for cells, we compared the cosine similarity between prediction and full batch at a high dimensional level. The histogram generally showed a high similarity score which further consolidates our down-sampling / prediction is robust to dense datasets.

In other scenarios when GPU resources are not available, users could run UniTVelo on CPU as well by setting `config.GPU = -1`. This also means higher peak memory usage and running time, the algorithm required 17GB memory and 9.0 hours for erythroid human dataset, 37k cells.

Overall, we provided effective trade-off approaches for users to analyse datasets with different sizes or configurations, while we agreed that the sampling method itself is an open challenge and we leave it to the future for a systematic comparison, especially when huge datasets are routinely used. We will monitor the feedbacks from users to further consider more sophisticated methods for running time optimization in future.

Minor comments:

The description of the criteria to select from two modes remains vague. I would suggest the authors add more specific guidelines. I understand that the users have implemented a utility script to choose the model. Maybe explicitly explain the concrete criteria the users had implemented. The authors also suggest running both modes. However, it would be difficult for users to judge results from which mode would be better. In addition, the running cost (time and memory) for running two modes will be prohibitive. Therefore, selecting a proper mode is not trivial and would require more explicit guidelines for users to follow.

Response:

Thank you for the kind suggestions! We have updated the selection criteria and briefly explained the rationale behind the utility script (Choose the suitable modes under Methods section, p10).

Indeed, determining the suitable mode for a specific dataset is a non-trivial and an interesting topic to explore both for scVelo (deterministic / stochastic / dynamical mode), UniTVelo (unified-time / independent mode), or other algorithms, as gene regulations during the biological process and associated expression profiles (phase portraits) are complex. The basic differences lie in how we reconstruct the temporal relationships between unspliced and spliced mRNA reads. The unified-time mode and

independent mode represent two opposite ways of exploration and suit different scenarios (or genes' activities). Nevertheless, although we recommend using unified-time as default, there are a few genes whose phase portraits exhibit clear, comprehensive induction or repression phases, and those genes are better to be fitted individually (the independent mode) and vice versa.

To elaborate, there is a trade-off between using two modes based on the displayed expression profiles. We anticipate a more comprehensive method to be developed for RNA velocity which may calculate gene-specific time matrix for a subset of genes whilst aggregate the time information of other genes.

In our utility script, we provided a simple way of choosing two modes,

- (1) The script will first identify whether cell cycle exists,
 - Assume we do not know the explicit clustering results (which is quite common in early phases of scRNA-seq analysis). Therefore, from a data point of view, we observed that for cycle related datasets, number of cycle genes in both S and G2M phases which are highly variable are significantly higher than other datasets. This could be a potential way to identify cycle related datasets and independent mode is recommended.
 - Specifically, after selecting the HVGs (same as scVelo and Seurat). Datasets with number of cycle genes in either S or G2M phases higher than half (an adjustable hyper-parameter) of the gene lists (43 S genes & 54 G2M genes) are considered as datasets with cell cycle included.

- (2) Then the script will detect whether this is a dataset with sparse cell types,
 - Normally scRNA-seq data and the related velocity streamlines are visualized on embeddings like UMAP, which reflects the similarity of expression profiles of various cell types.
 - To elaborate, sparsity refers to a few cell clusters scattered around with no obvious connections with others, meaning the proportion of its neighbour cells belong to the same cluster should be quite high. For now, we define if there are more than 2 clusters with more than 95% of its neighbour cells are within the same cluster (both are adjustable hyper-parameter), we consider this is a sparse dataset.

Despite its simplicity, we found this strategy effectively aligns with the 10 datasets we are using that have diverse biological properties. On the other hand, we admit that it may not perfectly generalise to all scenarios and we anticipate this will be an interesting topic and may motivate more future studies to develop comprehensive methods to determine these two modes or to integrate them into one framework.

REVIEWERS' COMMENTS

Reviewer #2 (Remarks to the Author):

The authors have addressed my remaining concerns with a very detailed response and convincing evidence. I have no further comments and now recommend the acceptance of the manuscript.